# Perinatal Docosahexaenoic Acid Supplementation Improves Cognition and Alters Brain Functional Organization in Piglets

**DOI:** 10.3390/nu12072090

**Published:** 2020-07-15

**Authors:** Xi Fang, Wenwu Sun, Julie Jeon, Michael Azain, Holly Kinder, Jeongyoun Ahn, Hee Cheol Chung, Ryan S. Mote, Nikolay M. Filipov, Qun Zhao, Srujana Rayalam, Hea Jin Park

**Affiliations:** 1Department of Foods and Nutrition, College of Family and Consumer Sciences, University of Georgia, Athens, GA 30602, USA; xi.fang25@uga.edu (X.F.); Julie.jeon@uga.edu (J.J.); 2Department of Physics and Astronomy, Franklin College of Arts and Sciences, University of Georgia, Athens, GA 30602, USA; ws40404@uga.edu (W.S.); qunzhao@uga.edu (Q.Z.); 3Department of Animal and Dairy Science, College of Agricultural & Environmental Sciences, University of Georgia, Athens, GA 30602, USA; mazain@uga.edu (M.A.); hollyk17@uga.edu (H.K.); 4Department of Statistics, Franklin College of Arts and Sciences, University of Georgia, Athens, GA 30602, USA; jyahn@uga.edu (J.A.); heecheol.chung@uga.edu (H.C.C.); 5Department of Physiology and Pharmacology, College of Veterinary Medicine, University of Georgia, Athens, GA 30602, USA; ryan.mote25@uga.edu (R.S.M.); filipov@uga.edu (N.M.F.); 6Department of Pharmaceutical Sciences, Philadelphia College of Osteopathic Medicine, Suwanee, GA 30024, USA

**Keywords:** omega-3 fatty acids, maternal nutrition, cognition, resting state network, brain development

## Abstract

Epidemiologic studies associate maternal docosahexaenoic acid (DHA)/DHA-containing seafood intake with enhanced cognitive development; although, it should be noted that interventional trials show inconsistent findings. We examined perinatal DHA supplementation on cognitive performance, brain anatomical and functional organization, and the brain monoamine neurotransmitter status of offspring using a piglet model. Sows were fed a control (CON) or a diet containing DHA (DHA) from late gestation throughout lactation. Piglets underwent an open field test (OFT), an object recognition test (ORT), and magnetic resonance imaging (MRI) to acquire anatomical, diffusion tensor imaging (DTI), and resting-state functional MRI (rs-fMRI) at weaning. Piglets from DHA-fed sows spent 95% more time sniffing the walls than CON in OFT and exhibited an elevated interest in the novel object in ORT, while CON piglets demonstrated no preference. Maternal DHA supplementation increased fiber length and tended to increase fractional anisotropy in the hippocampus of offspring than CON. DHA piglets exhibited increased functional connectivity in the cerebellar, visual, and default mode network and decreased activity in executive control and sensorimotor network compared to CON. The brain monoamine neurotransmitter levels did not differ in healthy offspring. Perinatal DHA supplementation may increase exploratory behaviors, improve recognition memory, enhance fiber tract integrity, and alter brain functional organization in offspring at weaning.

## 1. Introduction

Prenatal and early postnatal time represent critical periods of brain development sensitive to nutritional status [1]. Docosahexaenoic acid (DHA) is a n-3 long-chain polyunsaturated fatty acid (LCPUFA) commonly recommended during pregnancy and is widely added to the infant formula due to its suggested beneficial role in visual function and cognitive development [2,3]. DHA accounts for more than 40% of the total n-3 PUFA in the neuronal tissue [4], especially in the gray matter [5,6]. In the developing brain, the fastest DHA increase takes place during the third trimester of the gestation period, into the postnatal life [7,8], coinciding with the rapid myelination, dendritic outgrowth, and synaptogenesis [9], highlighting the importance of DHA during the critical window of neurodevelopment. Indeed, the perinatal deprivation of α-linolenic acid (ALA), the precursor of DHA, resulted in a 61% reduction in DHA concentration, a 65% decrease in serotonin (5-HT) levels in the prefrontal cortex (PFC) [10], and disturbed hippocampal (HC)-dependent learning and cognitive behaviors in rats compared to offspring born to ALA-sufficient animals [11]. Importantly, endogenous DHA synthesis is limited in mammals, and breast milk serves as a high bioavailable source of DHA to infants [12].

In humans, compelling evidence from observational studies supports the positive association between maternal fish intake and infant DHA status in neurocognitive development [13,14,15]. However, intervention trials supplementing DHA during gestation and/or lactation yielded mixed and conflicting findings in healthy term infants [16,17,18,19,20]. For example, the Avon Longitudinal Study of Parents and Children (ALSPAC) found that maternal seafood intake higher than 340 g/week is associated with better verbal intelligence quotient (IQ) and a decreased risk of suboptimal fine motor, social development, and communication scores of the children compared to those born to mothers who consumed less than 340 g seafood/week during pregnancy [14]. Nevertheless, a randomized control trial (RCT) in Australian women found that maternal supplementation of DHA (800 mg/day) during pregnancy did not influence cognitive and language development of infants at 18 months compared to the control group [21]. Similarly, DHA supplementation (220 mg/day) during gestation until 3 months after delivery did not influence the neurodevelopment of infants when they were evaluated at 18 months of age [22]; fish oil supplementation (400 mL/day) from the 28th week of gestation to 4 months lactation did not influence the visual and cognitive/psychomotor development of infants in their first year of life. These conflicting findings urge further exploration into the role of maternal DHA supplementation on the structural and functional organization of the developing brain in healthy subjects.

Pigs are a robust model for both nutritional and neurodevelopmental research in understanding nutritional programming in the human mother–infant dyad due to a myriad of similarities to humans in physiology, anatomy, pathology, and eating behavior [23,24,25,26]. Pigs have a gyrencephalic brain [27,28], unlike mice who are lissencephalic, a key architectural difference that has a direct correlation with brain connectivity and complexity [29,30,31,32]. The human and swine brain is composed of >60% white matter, while the white matter in the rodent brain is < 10% [33,34]. The anatomical similarities in different brain regions, such as the HC [35], subcortical and diencephalic nuclei [36], and cortical regions [37,38] between the pig and human brain have been illustrated. Furthermore, the pig brain exhibits a perinatal growth spurt much like human infants, during which the brain weight grows most rapidly from the last trimester of gestation to lactation [39,40]. Pig brain grows most rapidly from about 50 days pre- to 40 days postnatal [40]; the cerebrum presents two rapid growth periods between 80–100 days of conception and 6–26 days after birth [41]. In healthy human infants, the brain volume is around 25% of an adult’s brain at birth and grows rapidly to around 72 and 83% of an adult’s total brain volume at 1 year and 2 years after birth, respectively [42]. Longitudinal magnetic resonance imaging (MRI) analysis in pigs revealed that the total brain volume reaches 75% and 95% around 10.7 and 22.07 weeks after birth, respectively [43]. It is estimated that 1 week of piglet age is comparable to 1 month in human infants, and the 3-week-old piglets in this study are estimated to have brain development similar to that seen in human infants of around 3 months of age.

This is the first study, to our knowledge, that thoroughly examines the effect of perinatal DHA intake on the functional and cognitive development of the brain in piglets. The present study aims to investigate whether maternal supplementation of DHA during late gestation and lactation imparts advantages to cognitive development, fiber bundle maturation, anatomical and functional organization, and the monoamine neurotransmitter status of the developing brain in healthy offspring using a piglet model.

## 2. Materials and Methods

### 2.1. Animals and Study Design

The cross-bred commercial line of healthy pregnant sows (*n* = 8) were obtained from the University of Georgia Swine unit at approximately day 69 of gestation. Sows were assigned to the DHA group (DHA, *n* = 5) or the isocaloric control group (CON, *n* = 3) after accounting for the parity (5.38 ± 0.32) and body weight (BW, 239.7 ± 7.71 kg). After one week of acclimation in a temperature-controlled facility, sows were fed the corresponding diets from day 74 of gestation until delivery and throughout lactation. Approximately one week before expected farrowing, sows were transferred to individually house farrowing crates equipped with heat lamps. From each litter, 2–3 male and female piglets with a BW closest to the average BW were chosen within 24 h to be included in the study (*n* = 14 in CON, *n* = 20 in DHA). At weaning, piglets underwent behavior examinations, and a subset of piglets (*n* = 7 in both CON and DHA) underwent MRI acquisition and were sacrificed at approximately the postnatal day (PND) 20 for tissue collection (Figure 1). This study was conducted per the University of Georgia Institutional Animal Care and Use Committee guidelines (project code: A2018 04-003-Y2-A8).

### 2.2. Dietary Treatment

Pregnant sows were maintained on a gestation (2 kg/day) and lactation (ad libitum) diet primarily composed of corn with a total of 3300–3330 kcal/kg as recommended by National Research Council (NRC), 2012 [44]. Sows in the DHA group were supplemented with 75 mg/kg BW/day DHA as algae-produced DHASCO (contains 42.9% DHA, DSM Nutritional Products Inc., Pendergrass, GA, USA) by mixing the oil with the basal diet (Table 1). Poultry fat and corn are the primary sources of fat in the basal diet. The major fatty acids in the basal diet, based on the fatty acid composition of poultry fat and corn [45,46], are linoleic acid (C18:2n6, LA, approximately 46% in gestation diet and 43% in lactation diet), oleic acid (18:1, approximately 33% in gestation diet and 34% in lactation diet), and palmitic acid (16:0, approximately 13% in gestation and 15% in lactation diet). The current dosage of DHA was chosen based on previous findings that effectively enhances DHA level in sows and piglets and exerts beneficial health effects on the animals [47,48]. To determine whether this dose is safe, liver alanine aminotransferase was measured using a clinical chemistry analyzer that confirmed that the current dose did not cause hepatotoxicity. Sows in the CON group were on an isocaloric diet mixing the basal diet (Table 1) with safflower oil (Jedwards International Inc., Braintree, MA, USA), which contains predominantly LA [49] suggesting no additional intake of ALA in the control animals.

### 2.3. Colostrum Fatty Acid Composition

Colostrum was collected within 12 h postpartum from the functioning teats of the same position from all sows. The fatty acid concentration was measured using gas chromatography (Shimadzu, model 14 A, Tokyo, Japan) with a flame ionization detector as previously described [50]. Briefly, 2 mL colostrum samples were transmethylated according to the method of Park and Goins [51] and 2 mg of tridecanoic acid (C13:0) was added as an internal standard before processing. The upper hexane layer was collected, and residual water was removed by adding anhydrous sodium sulfate. Fatty acid methyl esters were separated on a Phenomenex, ZBWax Plus wide-bore capillary column (60 m × 0.53 mm, 1.00 μm film thickness; Phenomonex, Torrance, CA) with nitrogen as the carrier gas. The initial column temperature was 160 °C; the temperature was held for 10 min and increased at a rate of 5 °C/min until 220 °C. The injector temperature was 250 °C, and the detector temperature was 260 °C. Peaks were identified by comparison of the retention times of known standards (Nu-Chek Prep, Elysian, MN, USA).

### 2.4. Behavior Testing

In order to habituate piglets to human touch and novel environments, all piglets were handled and habituated on a daily basis from PND2 until the day of behavior tests.

#### 2.4.1. Open Field Test

As a measurement of ambulation and exploratory behaviors, piglets underwent the open field test (OFT) at approximately PND18 (PND 18.47 ± 0.097) in a 2.7 m × 2.7 m open arena lined with black mats. White curtains were hung around the arena to eliminate any visual distraction and the floor was sanitized with 70% ethanol between every trial to reduce olfactory bias. All piglets were exposed to the open arena for the first time. The piglets were individually introduced to the arena from the entry gate and allowed to explore for 10 min. Their exploratory behaviors (sniffing the wall), ambulation behaviors (mobile time, moving time), velocity, distance moved, and time spent in the center were recorded and measured by EthoVision video tracking software (Noldus, Wageningen, The Netherlands). The center zone size was designated as 0.9 m × 0.9 m in the center of the arena (Figure 2).

#### 2.4.2. Object Recognition Test

As a measurement of memory retention, piglets underwent the object recognition test (ORT), which has been successfully applied in pigs [52]. The ORT was comprised of two trials: a sample trial and a test trial. After habituation to the arena in OF, the piglet was reintroduced to the arena with two identical objects attached in the center of the open arena and was allowed to explore both objects for 10 min. The sample trial was immediately followed by a 10-min interphase delay, during which one object was replaced with a novel object. In the test trial, the piglet was introduced back to the arena and explored both the familiar and novel object for 10 min. All objects used in the study were cleaned thoroughly with water and ethanol between trials. The time that each piglet spent with each object in the sample and test trials were measured by Etho Vision (Noldus, Wageningen, the Netherlands). Proportional time was calculated as the ratio of time spent exploring the novel object to the total time exploring both objects in the sample and test trials.

### 2.5. MRI Acquisition

A subset of piglets (*n* = 14) underwent MRI imaging at approximately PND20 (PND 20.07 ± 0.31). Piglets were sedated with propofol for intubation (0.083–0.166 mL/kg, IV) and maintained under mild anesthesia with 1.5% isoflurane during scanning. MRI scanning including T1-weighted anatomical, diffusion tensor imaging (DTI), and resting-state fMRI (rs-fMRI) were conducted at the Bioimaging Research Center at the University of Georgia utilizing a 3.0 Tesla General Electric (GE) HDx scanner and a quadrature knee coil. Piglets were monitored throughout the scan by a veterinary technician. A 3D fast spoiled gradient echo sequence (repetition time (TR) = 5.5 s, echo time (TE) = 2.1 ms, flip angle (FA) = 9°, field of view (FOV) = 12.8 × 12.8 × 6.4 cm, slice thickness = 1 mm, acquired matrix = 256 × 256 × 112) was used to acquire T1-weighted anatomical data; a spin-echo echo-planar imaging (EPI) sequence (TR = 15.5 s, TE = min-full, FOV = 12.8 × 12.8 × 6.4 cm, acquired matrix = 64 × 64 × 32, and 30 diffusion weighted images using b = 1000 s/mm^2^) was used for DTI acquisition; and rs-fMRI was acquired by a gradient-echo EPI sequence (TR = 3 s, TE = 30 ms, FA = 80°, FOV = 12.8 × 12.8 × 6.4 cm, acquired matrix = 96 × 96 × 32, a total volume of 300 images).

### 2.6. MRI Analysis

#### 2.6.1. Anatomical and DTI Analysis

For anatomical MRI analysis, the individual brain was coregistered with a standard pig brain atlas [53] using the Statistical Parametric Mapping (SPM) toolbox. The percentage volume of 19 regions of interest were calculated using MATLAB (MATLAB R2018b, Natick, Massachusetts: The MathWorks Inc., Natick, MA, USA). For the DTI dataset, brain tissue was separated from the skull and other surrounding tissues by manual segmentation using a 3D Slicer 4.11.0, and tractography (mapping of nerve fiber tracts) was performed using the Tensor Toolkit (TTK) for tensor estimation and tensor tractography using the software MedInria (https://med.inria.fr/). A whole brain tractography was performed with fibers seeding from voxels with FA values greater than approximately double the whole-brain average and stopping at voxels with FA values less than approximately two-thirds of the whole-brain average. A more detailed, standard pig brain atlas [54] was then used to coregister each individual DTI dataset using the SPM toolbox in MATLAB. Measures of mean diffusivity (MD), fractional anisotropy (FA), and fiber length (FL) were obtained for the fibers intersecting with the region of interest from the atlas.

#### 2.6.2. Functional Connectivity Analysis

The rs-fMRI data (*n* = 7 for each group) were analyzed by using sparse dictionary learning (sDL) [55], a machine learning approach that has successfully detected brain functional connectivity [56]. Specifically, the fMRI time series was temporally concatenated to form a single matrix X, which was decomposed into two matrices, i.e., X = D × α, where D is the dictionary matrix aiming to learn temporal patterns from the concatenated time series (each column referred to as an atom), and α is a weight matrix that indicates the weights of learned features in the dataset. By using a back-propagation process, the dictionary was iteratively learned, and the corresponding α-matrix was used to generate functional connectivity maps, corresponding to each atom.

Pearson correlation (PC) was calculated by using the α-matrix and an atlas of piglet brain [56]. Three atoms with the highest PC coefficients were selected for the group analysis, separately for six resting state networks (RSNs), including the executive control network (ECN), cerebellum network (CERE), visual network (VIS), sensorimotor network (SMN), auditory network (AUD), and default mode network (DMN).

### 2.7. Animal Sacrifice and Tissue Collection

At weaning (PND 19.97 ± 0.14), all piglets were euthanized via CO_2_ asphyxiation. The brains were collected and coronally sectioned using a pig brain slicer (Zivic Instruments, Pittsburgh, PA, USA) and fresh tissues of HC and PFC were collected and immediately frozen in liquid nitrogen and stored in −80 °C for future analysis.

### 2.8. HPLC-ECD

Monoamine neurotransmitters, including dopamine (DA), serotonin (5-HT), and norepinephrine (NE) and their metabolites from PFC and HC were measured by the electrochemical detector for high-performance liquid chromatography (HPLC-ECD) as previously described [57,58]. Frozen tissue aliquots were mixed with 0.2 N perchloric acid (10 mg/100 μL), sonicated, and centrifuged. A total of 20 μL of the supernatant was injected into HPLC to determine: (1) DA and DA metabolites dihydroxyphenylacetic acid (DOPAC) and homovanillic acid (HVA); (2) 5-HT and its metabolite 5-hydroxyindoleacetic acid (5-HIAA); and (3) NE in brain tissues. The data were analyzed and presented as an ng of analyte per mg of tissue.

### 2.9. Statistical Analysis

Data expressed as mean ± S.E. were analyzed using R (version 3.6.0, R Foundation for Statistical Computing, Vienna, Austria) and GraphPad Prism (Version 7.00, GraphPad Software, Inc.; San Diego, CA, USA). The colostrum fatty acid composition from CON and DHA sows was analyzed using GraphPad Prism. For other data of piglets, a linear mixed-effect model was fitted to test the treatment while controlling the gender and maternal factors as fixed and random effects, respectively. The statistical significance of the treatment effect was determined based on *t*-statistics with the significance level of 0.05.

## 3. Results

### 3.1. Colostrum Fatty Acids Composition

The colostrum total lipids and lipid percentage did not differ in the two groups (*p* > 0.05, Table 2). The fatty acid profile of the colostrum from sows is comparable to that of the human colostrum, with high contents of palmitic acid (C16:0), oleic acid (C18:1), and linoleic acid (C18:2) [59]. Compared to CON, the DHA supplementation drastically increased the relative percentage levels of colostrum DHA (C22:6, 0.04 ± 0.01 and 3.36 ± 0.20 for CON and DHA, respectively, *p* < 0.0001) and other n-3 PUFA, including eicosapentaenoic acid (EPA, C20:5, 0.04 ± 0.02 and 0.48 ± 0.06 for CON and DHA, respectively, *p* = 0.001) and docosapentaenoic acid (DPA, C22:5, 0.17 ± 0.07 and 0.36 ± 0.04 for CON and DHA, respectively, *p* = 0.04) without influencing ALA level (C18:2, 0.83 ± 0.03 and 0.80 ± 0.07 for CON and DHA, *p* > 0.05) (Table 2). Concomitantly, the relative percentage levels of colostrum n-6 PUFA were markedly reduced in sows fed a DHA diet including LA (C18:2, 35.98 ± 2.03 and 27.55 ± 1.72 for CON and DHA, respectively, *p* = 0.02), dihomogammalinolenic acid (C20:3, 0.38 ± 0.05 and 0.23 ± 0.01 for CON and DHA, respectively, *p* = 0.01), and arachidonic acid (AA, C20:4, 1.16 ± 0.08 and 0.46 ± 0.01 for CON and DHA, respectively, *p* < 0.0001) (Table 2). Several other minor fatty acids, including lauric acid (C12:0), myristic acid (C14:0), myristoleic acid (C14:1), and palmitoleic acid (16:1), were also elevated in DHA-fed sows compared to CON (Table 2).

### 3.2. Piglet Bodyweight and Brain Weight

The piglets selected for this study were born with BW 1450.10 ± 68.84 g and grew to 5939.03 ± 288.00 g before weaning with an average daily weight gain of 223.48 ± 11.77 g/d and a final brain weight of 49.22 ± 0.63 g (averages of the two groups combined). Maternal supplementation of DHA and the isocaloric CON diet from late gestation to lactation did not influence the birth weight, final BW, weight gain, daily weight gain, and brain weight of the piglets at the end of the study (Table 3).

### 3.3. Behavior Testing

#### 3.3.1. Open Field Test

There was no statistically significant difference in the total distance moved, velocity, mobile time of any body part, and moving time of the body center point between CON and DHA piglets. This suggests that maternal DHA supplementation did not influence the locomotor function of healthy offspring (Figure 3A–D). Similarly, piglets from the two groups spent a comparable amount of time and similar visiting frequency in the center zone of the arena (Figure 3E–F). Interestingly, the piglets born to DHA-fed sows spent 95% more time sniffing the walls of the open arena compared to CON (*p* = 0.002, Figure 3G–H), demonstrating that the maternal DHA supplementation may enhance the inquisitive and exploratory behavior of the healthy offspring without affecting the normal development of motor function at weaning.

#### 3.3.2. Object Recognition Test

During the sample trial as explained in the methods section, all piglets spent similar time and frequency exploring the two identical objects with no preference (*p* > 0.05). In the test trial, male and female piglets born to CON-fed sows spent similar proportional time and frequency (Figure 4A–D) exploring the familiar and novel objects (*p* > 0.05). Male piglets from DHA-fed sows showed a tendency to spend more proportional time with the novel object (*p* = 0.060, Figure 4E), but not in proportional frequency (Figure 4F). The female piglets born to DHA-fed sows exhibited significantly more interest in the novel object than the familiar object, as they spent more proportional time (*p* = 0.032, Figure 4G) and proportional frequency (*p* = 0.008, Figure 4H) engaging with the novel object. This indicates that maternal DHA supplementation may improve the hippocampal-dependent short-term learning and memory of the offspring.

### 3.4. Structural MRI Analysis

The volumes of different brain regions in piglets born to CON and DHA-fed sows at weaning are presented in Table 4. At approximately PND 20, MRI volumetric assessments showed that piglet brains were composed of 63.4% cortex, 11.39% cerebellum, 2.76% thalamus, 1.84% HC, and 4.63% olfactory bulb and other subcortical regions. The maternal intake of DHA during late gestation and lactation did not pose any prominent change in the brain volumes of these healthy offspring. Although we found a statistically significant 0.0035% decrease in the thalamus, the clinical significance of this observation is unclear. In addition, the left and the right cortex tended to be slightly larger in male subjects compared to that of female subjects (left cortex: 31.2615% and 31.2565% for male and female, respectively; right cortex: 32.1408% and 32.1493% for male and female, respectively, *p* < 0.07). Overall, DHA supplementation during normal pregnancy and lactation did not result in significant changes in brain structural development in healthy offspring.

### 3.5. Hippocampal DTI Analysis

Axons insulated by a myelin sheath and firmly packed axonal bundles are critical for neuronal signal transduction and information processing, which may contribute to improved cognitive performance. Whole-brain tractography was performed in piglets born to CON- and DHA-fed sows (Figure 5A1–A2). Tract-based DTI analysis showed that DHA supplementation during gestation and lactation did not influence MD in piglet HC (*p* > 0.05, Figure 5B1). However, there was a trend of higher FA values in piglets born to DHA-fed sows (0.21 ± 0.003 and 0.23 ± 0.005 for CON and DHA, respectively, *p* = 0.07, Figure 5B2). Additionally, we observed a significant increase in FL in the HC of piglets born to DHA-fed sows (27.02 ± 0.93 mm) compared to piglets born to CON-fed sows (19.84 ± 2.07 mm, *p* = 0.01, Figure 5B3).

### 3.6. Functional Connectivity Analysis

In order to test whether maternal supplementation of DHA influences the large-scale cortical networks, we examined neural network organizations in six identified RSNs (Table 5). Earlier, using sDL, we successfully detected these networks in piglets that resemble their counterparts in human brains [56]. PC analysis was performed using the sDL activation maps and RSN atlas [56]. The PC analysis results and representative brain activation maps are presented in Figure 6 and Figure 7. Perinatal DHA supplementation resulted in an 8.7% increase in functional connectivity within the CERE (r = 0.3818 and 0.4152 for CON and DHA, respectively, Figure 6B/Figure 7B), 5.2% enhanced connectivity within the VIS (r = 0.3915 and 0.4120 for CON and DHA, respectively, Figure 6C/Figure 7C), 9.8% increase within the DMN (r = 0.3048 and 0.3346 for CON and DHA, respectively, Figure 6F/Figure 7F) and a minor increase within the AUD (r = 0.2523 and 0.2526 for CON and DHA, respectively, Figure 6E/Figure 7E) compared to CON piglets. In addition, piglets born to DHA-fed sows showed a 7.1% decrease in the ECN (r = 0.5217 and 0.4849 for CON and DHA, respectively, Figure 6A/Figure 7A) and an 8.9% decrease in functional connectivity in the SMN compared to that of CON piglets (r = 0.3149 and 0.4120 for CON and DHA, respectively, Figure 6D/Figure 7D).

### 3.7. Monoamine Neurotransmitters

In order to determine if the effects of DHA is through the regulation of monoamine neurotransmission, we measured key monoamine neurotransmitters and their metabolites using HPLC (Table 6). In the PFC, the levels of DA, 5-HT, NE and their metabolites did not differ between piglets born to CON and DHA-fed sows (*p* > 0.05). Piglets from DHA-fed sows had 32.94% higher DA, 30.48% higher 5-HT, and 45.83% higher NE in the HC relative to that of CON piglets; however, none of these reached statistical significance, likely due to the small sample size. Metabolites of DA and 5-HT in HC were not different between the two groups.

## 4. Discussion

DHA is commonly recommended for pregnant women and is widely added in infant formula for its potential benefits in visual and brain functions [60,61]. However, interventional trials have yielded conflicting neurocognitive outcomes. In this study, we found that maternal DHA supplementation increased exploratory behaviors, short-term object recognition memory, fiber length in the HC, and enhanced functional connectivity in key brain networks of healthy offspring.

We found that DHA supplementation during late gestation and lactation resulted in a remarkable 80-fold increase in the colostrum DHA concentration along with the other n-3 PUFA without influencing the ALA level. This is in agreement with the previous findings from humans and pigs that maternal dietary intake of DHA sufficiently modified breast milk DHA and other n-3 PUFAs, such as EPA (C20:5) and DPA (C22:5) levels, while ALA levels remained relatively stable [62,63,64,65,66]. Although we did not measure plasma DHA level in the offspring, other studies have shown that supplementing lactating mothers with DHA increased infant plasma and erythrocytes phospholipid DHA levels [64,65]. Meanwhile, breastmilk n-6 PUFAs, such as LA (C18:2) and AA (C20:4), were decreased due to dietary DHA supplementation in accordance with previous findings in humans, suggesting that reduced levels of these n-6 PUFAs in breast milk and a decreased n-6/n-3 ratio in infant plasma by DHA supplementation during pregnancy and lactation are likely due to the competitive incorporation of these two PUFAs into the plasma membrane [66,67].

Piglets born to DHA-fed sows demonstrated better cognitive performance in the behavioral testing. The OFT provides a simple and general measure of motor function and exploratory behaviors [68] and the ORT measures short-term object recognition memory, which is at least partly hippocampus dependent [69] in an enclosed and undisturbed setting. Previous studies demonstrated that selective hippocampal lesion led to impaired object recognition memory in both human and non-human primates [70,71], indicating a critical role of hippocampus in the short-term recognition memory. Additionally, the development of memory ability is suggested to be dependent partly on the progressive development of the hippocampus in a sequence of novelty preference, cognitive recall, flexible memory, and source memory [72]. Healthy offspring from DHA-fed and CON-fed sows exhibited similar locomotor functions. Interestingly, piglets from DHA-fed sows demonstrated more exploratory behaviors as they spent more time sniffing the walls of the open arena. Dietary intake of n-3-PUFA was found to be associated with increased exploratory behaviors in other models [73,74]. This may be due to decreased stress and anxiety levels and/or increased visual function of the piglets from the DHA group. In rodents, a low intake of n-3 PUFA reduced exploratory behaviors in young animals and dietary DHA decreases stress and anxiety levels [74,75,76]. In pigs, postnatal DHA deprivation depleted frontal cortex DHA and increased fear/anxiety-like behavior associated with brain DA levels [77]. The improved recognition memory formation assessed by ORT is in agreement with previous findings in which preterm piglets supplemented with DHA had increased recognition memory compared to the control [78]. The fact that the control piglets did not have a preference for the novel object at this stage indicates that a regulatory recommendation for DHA supplementation during neurodevelopment is worth considering. Nevertheless, more data and studies are needed as the memory-improving effect of DHA is widely seen in young adults [79] but not consistently observed in infants [20,80,81]. It is worth noting that these human trials commonly used the Fagan Test of Infant Intelligence to assess the recognition memory with only visual stimuli [82]. In comparison, the ORT in animals provides the subjects an opportunity to explore the objects with physical contact. The enhanced exploratory behavior and memory may also be related to the improved visual acuity due to DHA treatment. Human infants fed with a DHA-enriched diet showed accelerated development and maturation of the visual system and better visual function [83,84]. We also found increased functional connectivity within the visual network in these piglets, suggesting that the increased curiosity to the surrounding environment and higher engagement during object recognition may be partly due to the advanced development of the visual system.

MRI such as structural MRI, DTI, and rs-fMRI is increasingly used as a powerful tool to evaluate the role of DHA in the cognitive development of preterm infants. In our study, the perinatal DHA supplementation increased the myelination of axonal bundles in piglets. DTI measures brain microstructure and has been largely used to assess white matter integrity indicated by a reduced FA value [85,86]. Children born with very low birth weight had lower FA values in internal and external capsules, corpus callosum, and inferior and superior fasciculus than the control group, which was associated with visual-motor deficits and lower IQ scores [87]. A lower FA score was found to be associated with language processing, reading skills, and attention and anxiety behaviors later in life of preterm-born children, and such relation was not observed in full-term infants [88,89]. Breastfed infants have higher brain DHA concentrations [90], and preterm infants fed exclusively with breastmilk demonstrated greater structural connectivity and higher FA in major white matter fasciculi compared to non-exclusively breastfed infants [91]. Moreover, postnatal DHA supplementation to premature infants for 9 weeks showed a trend of a higher FA value in the corpus callosum related to control preterm infants at 8 years of age [92]. In this study, DTI analysis revealed an increased fiber tract length and a trend of increased FA value, which is in agreement with the previous findings in which higher DTI indices in the HC was seen in DHA-fed preterm pigs [78].

Resting-state fMRI assesses intrinsic functional connectivity between different neural networks by measuring fluctuations in blood oxygen level-dependent (BOLD) signal while the subject remains at a resting state without doing any cognitive task [93]. To the best of the authors’ knowledge, this is the first study investigating the intrinsic RSN changes due to maternal intake of DHA during gestation and lactation. CERE is associated with various fundamental functions, including motor coordination, visuomotor learning, executive function, and memory formation [94,95]. Higher activity of CERE due to perinatal DHA supplementation is likely to benefit the visual guidance of movement and working memory coinciding with our observations in the behavioral testing. DMN represents the intrinsic and spontaneous neuronal activity associated with internal thought processes and is deactivated during cognitively demanding tasks [96,97]. Failure of the DMN deactivation is associated with cognitive abnormities [98,99,100]. While it is unclear why perinatal DHA supplementation increased DMN activity in piglets, the intervention is likely to pose long-term influences on the cognitive processing and neurodegenerative processes [101]. Further investigations of maternal DHA supplementation on DMN activity comparing resting and task-based fMRI in infants will be of interest.

An increased VIS activity with DHA supplementation may indicate the accelerated maturation of the visual system, which underlies the improved visual acuity observed in human trials [83,84,102]. Term infants are born with adult-like VIS and SMN networks, suggesting prenatal development within these two domains [103]. We also found a decreased activity within SMN in piglets born to DHA-fed sows, which may suggest that DHA prevents the activation of SMN at the resting state. Interestingly, in human infants, the rapid development of SMN may occur earlier than the VIS [104], and the within-network connectivity of SMN manifests an age-related decrease during the first two years [103]; this may indicate an increased synaptic and axonal pruning within the sensory and motor cortices and a shift of functional organization towards more specialized cortical networks [105]. Thus, a decreased functional connectivity within SMN may suggest an enhanced development and maturation of the network that might have resulted from maternal DHA supplementation. Our findings suggest that perinatal DHA supplementation may alter brain functional organization of the offspring and support rs-fMRI as a sensitive tool to assess DHA status on brain functional connectivity and cortical organization in healthy piglets.

Monoamine neurotransmitters are critical neurochemicals for the proper function of learning, memory, and emotions [106]. Dietary *n* = 3-PUFA deficiency drastically decreased DA level and increased the 5-HT2 receptor in the PFC of young animals [107]. In piglets, postnatal DHA and AA supplementation to ALA/LA-deficient animals increased DA, 5-HT, and NE in the frontal cortex [108]. Perinatal ALA deficiency decreased tyrosine hydroxylase level, the rate-limiting enzyme for DA synthesis, in the substantia nigra and ventral tegmental area of the dopaminergic pathway [109] and increased 5-HT turnover in the PFC in rats [10]. We found that during a healthy pregnancy, maternal DHA supplementation resulted in an insignificant increase in DA, 5-HT, and NE in the HC of the offspring. It will be of interest to determine whether perinatal DHA intake influences neurotransmitters with a larger sample size.

This is the first study to provide evidence for improvement in cognition and brain development with perinatal DHA supplementation in piglets. However, there are a few limitations to this study. A small sample size, though statistically enough, may have limited our power to detect the true effect of perinatal DHA supplementation in piglets and increased the error due to the random effects, including sow effects. The maternal factor was controlled in statistical analysis by including sow as a random effect using the mixed-effect linear model. Additionally, the present study used safflower oil as a control oil, which is high in n-6 PUFA. Thus, the observed changes may be ascribed in part to the high n-6 intake in the control animals. Although n-6-rich components like safflower oil are commonly used in the control diets to match the caloric intake between the control and test groups, safflower oil is low in oleic acid, which is the main fatty acid of myelin and could potentially influence myelination and brain functional connectivity during development [110]. Finally, the current study measured fMRI when the animals were at resting state. While it provided valuable insights into the resting-state networks influenced by the treatment, it is challenging to interpret due to its novel application in infant piglets. Future studies integrating both resting-state and task-based fMRI in a larger sample size would be of interest.

## 5. Conclusions

In conclusion, the present data provides preclinical support that the maternal DHA supplementation during late gestation and lactation may increase exploratory behavior, improve memory function, enhance fiber tract integrity, and alter brain functional organization of offspring at weaning without affecting the volume of major brain structures.

## Figures and Tables

**Figure 1 nutrients-12-02090-f001:**
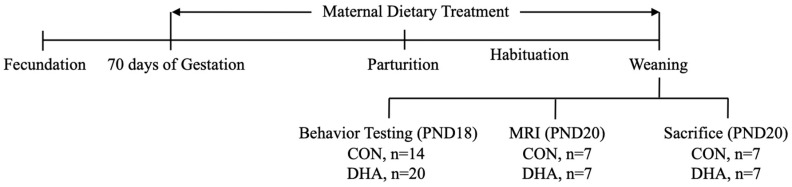
Study design and experimental timeline. CON: control; DHA: docosahexaenoic acid.

**Figure 2 nutrients-12-02090-f002:**
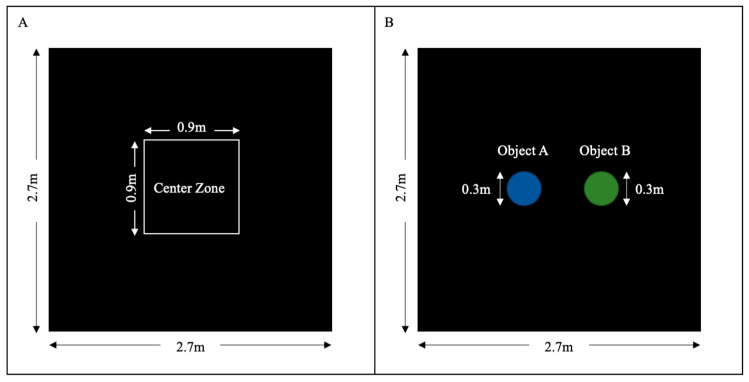
Schematic illustration of the testing arena for open field test (**A**) and the object recognition test (**B**).

**Figure 3 nutrients-12-02090-f003:**
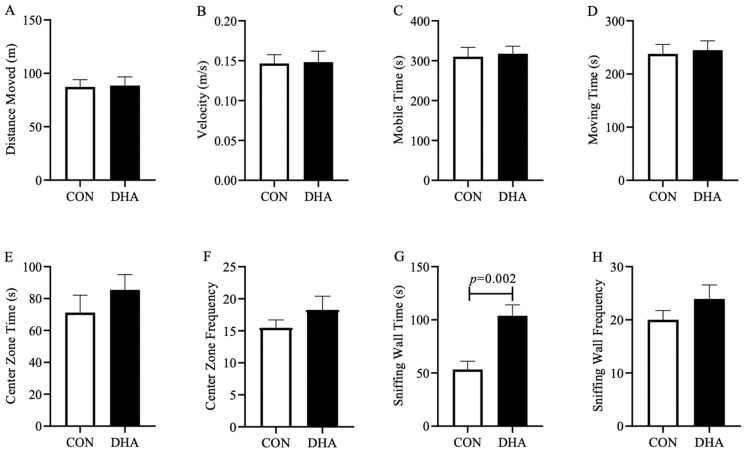
Measures of activity in the open field test for piglets born to sows fed a CON (*n* = 14) or DHA (*n* = 20) diet at weaning. (**A**) distance moved; (**B**) velocity; (**C**) mobile time of any part of the body; (**D**) moving time of the center point; (**E**) time spent in the center zone; (**F**) frequency visit the center zone; (**G**) sniffing wall time; (**H**) sniffing wall frequency. A linear mixed-effect model was used to control for gender (fixed) and maternal (random) effects. Abbreviations: CON: control; DHA: docosahexaenoic acid. *p* values higher than 0.1 are not shown on the graph.

**Figure 4 nutrients-12-02090-f004:**
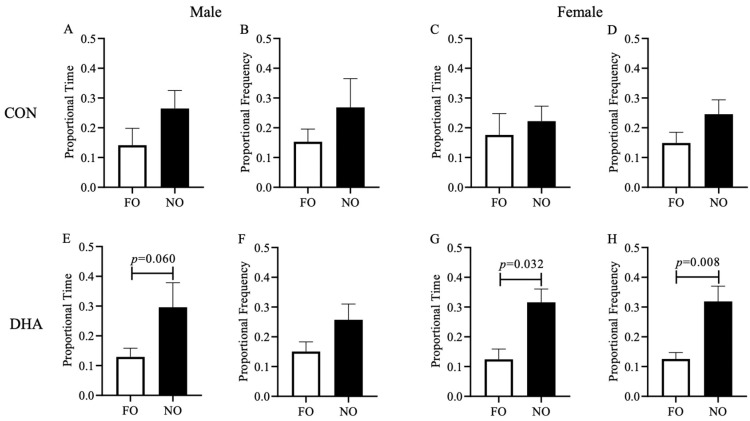
Measures of activity in the test trial of novel object recognition test for piglets born to sows fed a CON (*n* = 14) or DHA (*n* = 20) diet at weaning. Figures depicting proportional time and visit frequency on the familiar and novel objects. (**A**) proportional time of control male piglets; (**B**) proportional frequency of control male piglets; (**C**) proportional time of control female piglets; (**D**) proportional frequency of control female piglets; (**E**) proportional time of DHA male piglets; (**F**) proportional frequency of DHA male piglets; (**G**) proportional time of DHA female piglets; (**H**) proportional frequency of DHA female piglets; a linear mixed-effect model was used to control for gender (fixed) and maternal (random) effects. Abbreviations: FO: familiar object; NO: novel object. *p* values higher than 0.1 are not shown on the graph.

**Figure 5 nutrients-12-02090-f005:**
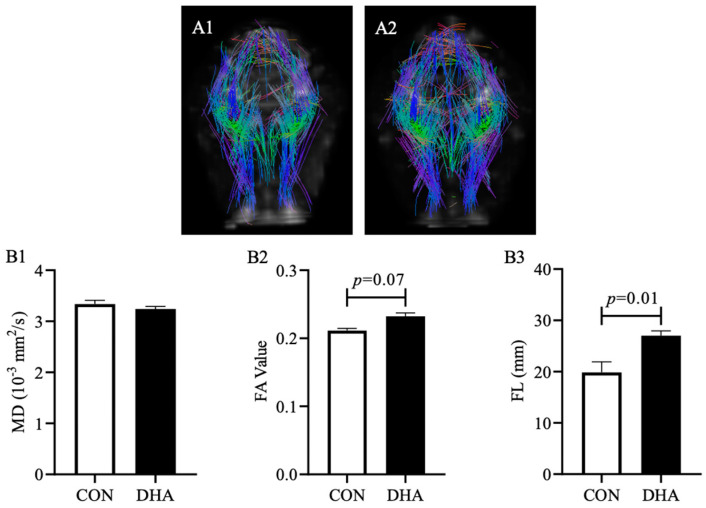
Diffusion tensor imaging analysis. Representative whole brain tractography of piglets born to sows fed a CON (**A1**, *n* = 7) or DHA (**A2**, *n* = 7) diet during late gestation and lactation. Mean diffusivity (**B1**), fractional anisotropy (**B2**), and fiber length (**B3**) within the hippocampus of piglets. A linear mixed-effect model was used to control for gender (fixed) and maternal (random) effects. Abbreviations: CON: control; DHA: docosahexaenoic acid; MD: mean diffusivity; FA: fractional anisotropy; FL: fiber length. *p* values higher than 0.1 are not shown on the graph.

**Figure 6 nutrients-12-02090-f006:**
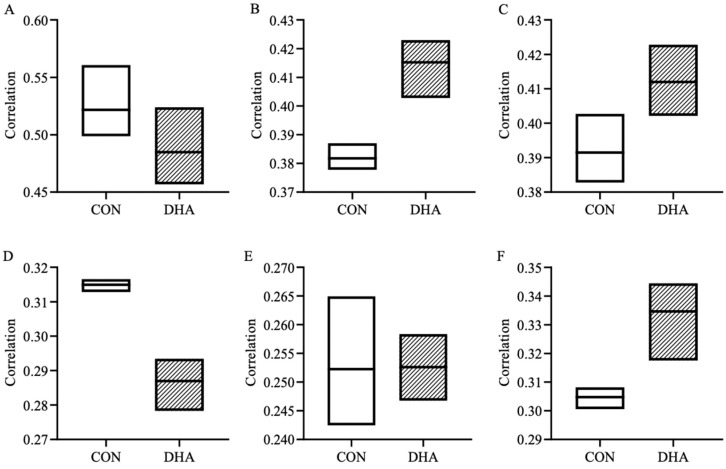
Group analysis of functional connectivity from rs-fMRI scan in six resting state networks, including the ECN (**A**), CERE (**B**), VIS (**C**), SMN (**D**), AUD (**E**), and DMN (**F**), in piglets born to sows fed a CON (*n* = 7) or DHA (*n* = 7) diet. The graphs show the correlation coefficients from the best three activation maps. Abbreviations: CON: control; DHA: docosahexaenoic acid. RSN: resting state network; ECN: executive control network; CERE: cerebellar network; VIS: visual network; SMN: sensorimotor network; AUD: auditory network; DMN: default mode network.

**Figure 7 nutrients-12-02090-f007:**
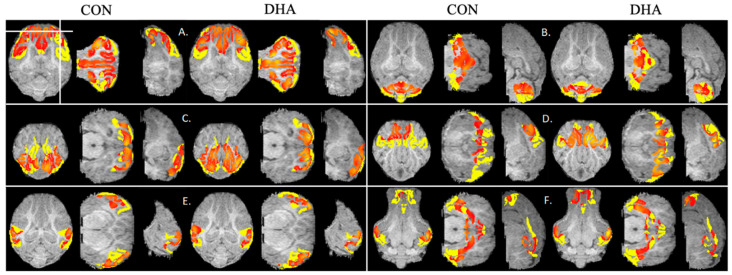
Color-coded regions illustrating activation within six RSNs, including ECN (**A**), CERE (**B**), VIS (**C**), SMN (**D**), AUD (**E**), and DMN (**F**), in piglets born to sows fed a CON (*n* = 7) or DHA (*n* = 7) diet. Each session contains activation maps of CON (left) and DHA (right) piglets in axial, coronal, and sagittal planes. Horizontal and vertical bars on the axial image at the upper-left corner indicate locations of the coronal and sagittal images. Yellow patters are RSN atlases and orange patters are activations. Abbreviations: CON: control; DHA: docosahexaenoic acid. RSN: resting state network; ECN: executive control network; CERE: cerebellar network; VIS: visual network; SMN: sensorimotor network; AUD: auditory network; DMN: default mode network.

**Table 1 nutrients-12-02090-t001:** Diet composition of the basal diet during gestation and lactation.

Ingredients, g/kg	Gestation Diet ^1^	Lactation Diet ^1^
Corn	535.4	389.6
Soybean Meal	32.3	172.3
Distillers Dried Grains with Solubles	400.0	400.0
Fat		4.6
Dicalcium Phosphate	2.8	
Limestone	17.4	21.6
Salt	3.5	3.5
Vitamin pre-mix ^2^	2.5	2.5
Trace Mineral pre-mix ^3^	1.5	1.5
Sow Vitamin pre-mix ^4^	2.5	2.5
L-Lysine HCl	2.1	2.0
Total	1000.0	1000.0
Calculated Analysis		
Crude Protein, %	17.4	22.4
Metabolizable Energy, kcal/kg	3330	3300
Crude Fiber, %	4.18	5.10
Ether Extract, %	6.23	6.14
Lysine, %	0.70	1.10
TSAA, %	0.68	0.86
Threonine, %	0.64	0.92
Tryptophan, %	0.15	0.25
Ca, %	0.79	0.90
Total *P*, %	0.52	0.57
Available *P*, %	0.29	0.40

^1^ Basal diet was supplemented with 75 mg/kg BW/day DHA as DHASCO or an equivalent amount of safflower oil daily in the DHA group and CON group, respectively. ^2^ Supplied per kg of premix: vitamin A 4400 IU; vitamin D 660,000 IU; vitamin E 17,600 IU; vitamin K 1760 IU; riboflavin 3960 mg; niacin 22,000 mg; vitamin B12 17,600 μg. ^3^ Supplied per kg of premix: iron 110,000 mg; copper 11,000 mg; manganese 26,400 mg; zinc 110,000 mg; iodine 198 mg; selenium 198 mg. ^4^ Supplied per kg of premix: biotin, 88 mg; choline, 220.5 g; folic acid, 661.5 mg; pyridoxine, 1.98g; vitamin E8,882 IU.

**Table 2 nutrients-12-02090-t002:** Colostrum fatty acid composition from sows fed with/without DHA during late gestation and lactation.

Compound	Control (*n* = 3)	DHA (*n* = 5)	*p*-Value
lipid	3.90 ± 0.90	3.83 ± 0.55	0.94
C12:0	0.04 ± 0.00	0.09 ± 0.00	0.01
C14:0	1.23 ± 0.09	2.29 ± 0.17	0.004
C14:1	0.02 ± 0.00	0.07 ± 0.01	0.01
C15:0	0.11 ± 0.01	0.11 ± 0.01	0.69
C16:0	18.31 ± 0.69	19.50 ± 0.50	0.20
C16:1	3.25 ± 0.07	4.43 ± 0.34	0.04
C17:0	0.30 ± 0.01	0.24 ± 0.01	0.01
C17:1	0.23 ± 0.01	0.24 ± 0.02	0.90
C18:0	4.70 ± 0.52	4.30 ± 0.32	0.51
C18:1	29.80 ± 0.81	32.18 ± 1.42	0.19
C18:2	35.98 ± 2.03	27.55 ± 1.72	0.02
C18:3n-6	0.64 ± 0.08	0.39 ± 0.05	0.03
C18:3n-3	0.83 ± 0.03	0.80 ± 0.07	0.78
C20:0	0.18 ± 0.07	0.14 ± 0.03	0.65
C20:1	0.34 ± 0.03	0.37 ± 0.02	0.45
C20:2	0.74 ± 0.04	0.66 ± 0.03	0.14
C20:3	0.38 ± 0.05	0.23 ± 0.01	0.01
C20:4	1.16 ± 0.08	0.46 ± 0.01	< 0.0001
C20:5	0.04 ± 0.02	0.48 ± 0.06	0.001
C22:2	0.59 ± 0.35	0.64 ± 0.04	0.84
C22:5	0.17 ± 0.07	0.36 ± 0.04	0.04
C22:6	0.04 ± 0.01	3.36 ± 0.20	< 0.0001

Relative percent of each fatty acid is shown. The data are presented as means ± S.E. Abbreviations: DHA: docosahexaenoic acid.

**Table 3 nutrients-12-02090-t003:** Body weight and brain weight of piglets at weaning.

Parameter	Control (*n* = 14)	DHA (*n* = 20)	*p*-Value
Birth weight (g)	1539.00 ± 75.56	1389.00 ± 103.10	0.67
Final weight (g)	6134.00 ± 353.60	5790.00 ± 436.40	0.73
Weight gain (g)	4528.00 ± 338.00	4333.00 ± 308.70	0.78
Daily weight gain (g/day)	229.30 ± 16.87	219.80 ± 16.28	0.77
Brain weight (g)	49.28 ± 0.69	49.17 ± 0.97	0.77

The data are presented as means ± S.E. A linear mixed-effect model was used to control for gender (fixed) and maternal (random) effects. Abbreviations: DHA: docosahexaenoic acid.

**Table 4 nutrients-12-02090-t004:** Structural magnetic resonance imaging (MRI) analysis of nineteen brain regions of piglets at weaning.

Brain Region	Control (*n* = 7)	DHA (*n* = 7)	*p*-Value
Caudate	0.860 ± 0.001	0.861 ± 0.001	0.46
Cerebellum	11.389 ± 0.002	11.388 ± 0.001	0.71
Left Cortex	31.262 ± 0.002	31.257 ± 0.002	0.09
Right Cortex	32.143 ± 0.002	32.144 ± 0.001	0.54
Lateral Ventricle	1.012 ± 0.002	1.014 ± 0.0001	0.95
Third Ventricle	0.106 ± 0.0003	0.105 ± 0.0005	0.77
Cerebral Aqueduct	0.074 ± 0.0003	0.074 ± 0.0002	0.63
Fourth Ventricle	0.099 ± 0.0005	0.100 ± 0.0002	0.33
Left Hippocampus	0.914 ± 0.0005	0.916 ± 0.0006	0.15
Right Hippocampus	0.925 ± 0.001	0.926 ± 0.001	0.50
Medulla	3.364 ± 0.002	3.3620 ± 0.001	0.54
Midbrain	3.414 ± 0.001	3.412 ± 0.001	0.31
Pons	2.154 ± 0.001	2.155 ± 0.001	0.44
Putamen and Globus Pallidus	0.731 ± 0.001	0.731 ± 0.0004	0.78
Hypothalamus	0.501 ± 0.001	0.502 ± 0.001	0.13
Thalamus	2.771 ± 0.001	2.767 ± 0.001	0.04
Olfactory Bulb	4.632 ± 0.001	4.634 ± 0.002	0.48
Corpus Callosum	0.776 ± 0.002	0.775 ± 0.001	0.86
Internal Capsule	2.875 ± 0.002	2.877 ± 0.001	0.55
Total (voxel)	411301 ± 11.94	411342 ± 25.47	0.23

Percentage volume of different brain regions is shown. The data are presented as means ± S.E. A linear mixed-effect model was used to control for gender (fixed) and maternal (random) effects. Abbreviations: DHA: docosahexaenoic acid.

**Table 5 nutrients-12-02090-t005:** Resting-state network included in the analysis and their regions of interest.

Resting-State Network	Locations of Regions of Interest
Executive control network (ECN)	Primary somatosensory cortex
	Dorsolateral prefrontal cortex
	Anterior prefrontal cortex
	Orbitofrontal cortex
	Insular cortex
	Ventral anterior cingulate cortex
	Dorsal anterior cingulate cortex
Cerebellar network (CERE)	Cerebellum
Visual network (VIS)	Primary visual cortex
	Secondary visual cortex
	Associative visual cortex
Sensorimotor network (SMN)	Primary motor cortex
	Somatosensory association cortex
	Premotor cortex
Auditory network (AUD)	Superior temporal gyrus
	Auditory cortex
Default mode network (DMN)	Hippocampus
	Anterior prefrontal cortex
	Orbitofrontal cortex
	Inferior temporal gyrus
	Ventral posterior cingulate cortex
	Retrosplenial cingular cortex
	Dorsal posterior cingular cortex
	Anterior entorhinal cortex
	Parahippocampal cortex

**Table 6 nutrients-12-02090-t006:** Concentrations of monoamines and their metabolites in the PFC and HC in piglets at weaning.

Brain Regions	Neurochemicals	Control (*n* = 7)	DHA (*n* = 7)	*p-*Value
PFC	DA	3.463 ± 0.390	3.551 ± 0.454	0.85
	DOPAC	0.090 ± 0.012	0.078 ± 0.015	0.64
	HVA	0.121 ± 0.015	0.092 ± 0.009	0.61
	5-HT	0.174 ± 0.022	0.124 ± 0.030	0.08
	5-HIAA	0.067 ± 0.010	0.050 ± 0.010	0.25
	NE	0.168 ± 0.022	0.163 ± 0.039	0.89
HC	DA	2.459 ± 0.181	3.269 ± 0.371	0.14
	HVA	0.170 ± 0.035	0.155 ± 0.024	0.99
	5-HT	0.105 ± 0.017	0.137 ± 0.024	0.26
	5-HIAA	0.096 ± 0.004	0.113 ± 0.019	0.36
	NE	0.072 ± 0.008	0.105 ± 0.014	0.11

The data are presented as means ± S.E; unit: ng/mg protein. Abbreviations: DHA: docosahexaenoic acid; PFC: prefrontal cortex; HC: hippocampus; DA: dopamine; DOPAC: dihydroxyphenylacetic acid; HVA: homovanillic acid; 5-HT: serotonin; 5-HIAA: 5-hydroxyindoleacetic acid; NE: norepinephrine; A linear mixed-effect model was used to control for gender (fixed) and maternal (random) effects.

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
