# Peer review of "Perinatal Docosahexaenoic Acid Supplementation Improves Cognition and Alters Brain Functional Organization in Piglets"

_nutrients, 2020, doi:10.3390/nu12072090_

Round 1

Reviewer 1 Report

This study aims at assessing how perinatal DHA supplementation affects cognitive performance, brain organization, and monoamine neurotransmitter status of offspring using a piglet model.

The manuscript is sound and well-written. I only have minor comments:

1- Both males and females were analyzed. However, in the results/graphs, it is not clear if data are from males, or females or a mix of both? I would suggest splitting male and female data in all Tables and Figures, as it was done in Figure 2. For instance, for structural MRI analyses, the authors report on a male/female comparison in the Result section, yet, this does not appear in Table 4.

2- The Inter-Trial Interval (ITI) is of 10 minutes in the Open-Field test. I am not familiar with the piglet model, yet in rodents, one would use an ITI of at least 30 minutes to an hour at that age (PND21), as the only way to actually test short term memory. Such a short ITI is usually used to reveal potential sensory alterations rather than memory deficits in piglets. This is a critical issue since the authors also found an increase in the visual system activity in their MRI data. please comment.

3- The authors used n=14 controls and n=20 DHA-supplemented piglets. Why such a difference? Was this taken into account in the statistical analyses?

4- The role of the hippocampus in the ORT is still a matter of debate, except when the novel object is moved to another location, in which case spatial memory is required. Could the authors comment on this in the Discussion section and justify further their interest in the hippocampus in this study?

5- As stated earlier, I am not familiar with the piglet model, hence it is not clear to me what PND21 means for this species in terms of brain development. Could the authors add some information on that in the Introduction or Methods section? That being said, I fully acknowledge their attempt to explain their model in more detail to adapt to the readership.

Reviewer 2 Report

Fortification of infant formula with long chain polyunsaturated fatty acid (LCPUFA), namely arachidonic acid (ARA) and docosahexaenoic acid (DHA), has been recommended to provide levels of ARA and DHA found naturally in breast milk. Results of observational studies suggest that maternal consumption of fish and seafood containing DHA is associated with better infant neurocognitive development. However, data from randomized controlled trials supplementing mothers with DHA failed to demonstrate any significant impact on infantile cognitive endpoints. There is a limited understanding of the impact of maternal supplementation with DHA has on the structural organization of the developing brain. The authors use a piglet model to evaluate the structural changes in the brain of piglets of sows fed either DHA or a “control” diet not containing DHA. In addition to marginal behavioral differences between treatment groups, DHA exposure to piglets contributed to structural and functional brain changes compared to the “control” group. While by and large the manuscript is of adequate quality, the authors need to address some outstanding limitations of their study.

MAJOR concerns:

  1. The description of the maternal diets is lacking given that it is the major exposure variable in the current experiment. Specifically, the authors need to report the fatty acid profile of the background fat/oil in the base diet. As it stands there is no explicit detail regarding source of the fat/oil used in the base diet.

  1. If the fat/oil used in the base diet is from corn, as indirectly suggested by the authors, the use of the term “control” for the diet containing safflower oil is discouraged. This diet should more appropriately be described as a high n-6 PUFA diet or a n-3 PUFA deprived as the level of n-3 PUFA in the maternal diet would likely be insufficient. Review of the colostrum fatty acid profile indicates that the maternal diet is indeed very high in n-6 PUFA as the n-6 to n-3 ratio is approximately 38:1 in the “control” group. This is considerably higher than what is seen in global human milk fatty acid profiles (Yuhas, Pramuk and Lien, Lipids. 2006; 41:851-858).

  1. The authors must include a discussion of the limitations of their study.

Minor concerns:

  1. “n3” should be hyphenated to “n-3” and corrected throughout the manuscript.
  2. Table 1 is lacking units for some of the dietary components
  3. Table 2 the use of “%” for each fatty acid is redundant given the description accompanying the table.
  4. The authors should include a comment on the fatty acid profile sow colostrum compared to breastmilk or colostrum of human mothers from intervention studies.
  5. It would help the reader if sample sizes for each analysis accompanied each figure and table.

Reviewer 3 Report

This is an interesting study that thoroughly examines the effect of perinatal DHA intake for the cognitive function of piglets.

Major

1) Dietary fatty acids: Can the authors provide data on the fatty acid composition in all the diets used in the study? Is there any measure taken to prevent the oxidation of DHA in the diets?

2) Study design: Why are 5 sows allocated to the DHA group, while 3 were allocated to control, but not 4 and 4?

3) Behavioural assessments: It is not clear how many animals were used for each behavioural test. Are there any data excluded? I would suggest the authors to provide a figure summarising the timeline of behavioural experiments (showing orders of tests) and how many animals are used. Has power analysis been performed (will be good to be included in the response to reviewer, no need to include in the manuscript)? 

Minor

1) OFT: Can the authors provide a figure showing how centre zones are defined? By any chance has the wall sniffing time counted using software been verified using manual counting?

2) Figure 1 & 2: Can the authors include n number in the figure legends?

3) Can the authors comments if the the colostrum fatty acids composition of sows are comparable to humans? Has any human studies been done that contains the similar DHA composition in the breast milk after dietary supplement?
